# Curated mitochondrial genome reference database of state key protected wild mammal in China

Xia Huang[1,☯], Guihua Zhang[1,☯], Joseph D. Orkin[2,3,4], Shiyun Liu[1], Shan Jiang[1], Yinhui Zhao[5,6], Pengfei Fan[7], Lianghua Huang[8], Xiaoming Zhang[5,6], Xueyou Li[5], Song Li[9*], Kai He[1*]

1 Key Laboratory of Conservation and Application in Biodiversity of South China, School of Life Sciences, Guangzhou University, Guangzhou, China, 2 Département d'anthropologie, Université de Montréal, Montréal, Québec, Canada, 3 Département de sciences biologiques, Université de Montréal, Montréal, Québec, Canada, 4 IBE, Institute of Evolutionary Biology (UPF–CSIC), Department of Medicine and Life Sciences, Universitat Pompeu Fabra, Barcelona, Spain, 5 State Key Laboratory of Genetic Evolution and Animal Models, Kunming Institute of Zoology, Chinese Academy of Sciences, Kunming, China, 6 Yunnan Key Laboratory of Integrative Anthropology, Kunming, China, 7 School of Life Sciences, Sun Yat-sen University, Guangzhou, China, 8 Wildlife Forensic Science Service, Kunming, China, 9 Kunming Natural History Museum of Zoology, Kunming Institute of Zoology, Chinese Academy of Sciences, Kunming, China

☯ These authors contributed equally to this work.
* hekai@gzhu.edu.cn

## Abstract

Effective conservation of wild mammals necessitates accurate taxonomic classification and reliable genetic reference data. In China, the List of State Key Protected Wild Animals serves as a critical tool for species protection. However, taxonomic revisions and gaps in genetic data can impede its effectiveness. In this study, we updated the List of State Key Protected Wild Animals (2021) by incorporating recent taxonomic and distributional evidence, resulting in a refined list of 169 mammalian species that are protected. We identified 15 taxa lacking complete mitochondrial genome data and addressed this gap by generating 12 new mitogenomes for nine taxa using a combination of GenBank database mining and next-generation sequencing of museum specimens and fecal samples. These efforts led to the establishment of a curated mitochondrial genome reference database encompassing 164 species. Our analyses also uncovered taxonomic ambiguities in genera such as *Moschus* and *Naemorhedus*, and highlighted mislabeling issues within public genetic databases. This curated database enhances the accuracy of forensic species identification, supports biodiversity monitoring, and strengthens wildlife law enforcement. Our findings underscore the value of integrating historical specimens with mitogenomic approaches to advance wildlife conservation efforts.

**Data availability statement:** All relevant data are within the manuscript and its Supporting Information files.

**Funding:** This work was funded by the National Natural Science Foundation of China, Grant numbers: U23A20161 and 32170452. The funders had no role in study design, data collection and analysis, decision to publish, or preparation of the manuscript.

**Competing interests:** The authors have declared that no competing interests exist.

## Introduction

Wild mammals are integral to ecosystems, contributing to processes such as nutrient cycling, seed dispersal, pollination, and pest control [1]. However, climate change, habitat loss, fragmentation, and degradation, have led to significant population declines, pushing many species toward extinction [2–4]. In response to the accelerating biodiversity crisis, governments and conservation organizations have intensified efforts to protect vulnerable species, notably through the development and regular updating of species protection lists such as the International Union for Conservation of Nature (IUCN) Red List. To capture nation-specific conservation needs, the Chinese government has developed its national classification system: The List of State Key Protected Wild Animals (LSKPWA), which is updated approximately every five years. This list divides species into Class I and Class II based on their conservation urgency, legal status, and ecological importance, with Class I encompassing species at greater risk. The latest revision, published in 2021, expanded the list to include 980 species and eight categories, with 234 species and one category under Class I, and 746 species and seven categories under Class II (http://www.forestry.gov.cn/main/5461/20210205/122418860831352.html). In China, the legal penalties for the illegal hunting, killing, acquisition, transportation, or sale of wildlife vary depending on the species' protection status.

Despite efforts to update LSKPWA, continuous taxonomic revisions have led to discrepancies between the LSKPWA and current species classifications in China. For instance, mammalian taxonomy has undergone extensive changes in recent years [5]. These revisions include the elevation of subspecies to full species [6], confirmation of certain species' absence from China, and synonymization of species with existing taxa [7]. These taxonomic updates result in misalignments between species inventories and LSKPWA. For example, red pandas (*Ailurus fulgens*), which are listed as Class II in the 2021 LSKPWA, have been reclassified into two species: the Chinese red panda (*A. styani*) with a stable, widespread population and the Himalayan red panda (*A. fulgens*) with a restricted range and critically low population size [6]. Such misalignments can obscure conservation priorities and hinder effective law enforcement.

In addition to the misalignments, accurate species identification is further complicated by morphological similarities among certain species, despite differences in their conservation status. For instance, three species within the genus *Martes* are morphologically similar, but only one is listed under Class I, while the others fall under Class II. Distinguishing these species based solely on appearance is challenging, particularly for non-specialists or when dealing with incomplete specimens. Furthermore, confiscated wildlife materials often consist of partial remains (e.g., hides, bones, or frozen tissues) or processed products (e.g., musk, pangolin scales, ivory, or horn products), which lack diagnostic morphological features, necessitating molecular identification techniques [8,9].

DNA barcoding is a widely adopted tool for species identification in both ecological and forensic contexts [10,11]. Traditional barcoding typically uses short mitochondrial gene fragments, such as two ribosomal RNAs (12S rRNA and 16S rRNA),

cytochrome c oxidase subunit 1 *(COI)*, or cytochrome b *(CYTB)* [12,13]. While sufficient in most cases, these single-gene markers could be insufficient when closely related species exhibit low interspecific divergence. For instance, COI could not reliably distinguish between *Procapra przewalskii* and *P. gutturosa*, which exhibit only 0.5% genetic divergence in this gene [14]. In contrast, complete mitochondrial genome sequences provided clear phylogenetic resolution between the two species allowing for accurate identification [15]. This illustrates the need for complete mitochondrial genomes (mitogenomes) as a more comprehensive reference data.

Although GenBank hosts a growing collection of mitogenome sequences, gaps and inconsistencies remain. Our preliminary review revealed that at least 15 mammalian species or subspecies listed in China's LSKPWA still lack complete mitogenomes, including multiple primates and ungulates (S1 Table). Moreover, public databases commonly include misidentification or mislabeling [16,17], which, if used uncritically, can lead to erroneous species identifications and legal misinterpretations.

To address these challenges, this study aims to establish a curated mitochondrial genome reference database for the state key protected wild mammals in China. The objectives include updating the list of protected species to incorporate current taxonomic and distributional data, identifying gaps in existing mitogenome datasets, and using next-generation sequencing technologies to generate new sequences. This curated database will serve as a crucial tool for forensic investigations, enhancing the accuracy of species identification in cases of wildlife trafficking, and other legal matters related to conservation.

## Materials and methods

### Updating the list of state key protected wild mammals in China

To update the taxonomy of species in the 2021 LSKPWA, we integrate the latest expert consensus from key national publications [5,7] and the internationally recognized *Mammal Diversity Database v2.3* (https://zenodo.org/records/17033774; last update September 2, 2025), which is actively curated by the American Society of Mammalogists. Each revision was cross-referenced with the latest peer-reviewed publications and verified by relevant experts.

### Taxon sampling, mitogenome sequencing and assembly

We screened the updated species list against GenBank and identified ten species and five subspecies lacking mitogenomes (S1 Table). These included four primates, two carnivores, one lagomorph, and eight cetartiodactylans.

We collected 10 samples representing seven species, including feces, museum skin, and tissue samples (Table 1). Genomic DNA from museum skins and tissue samples was extracted using the DNeasy Blood & Tissue Kit (Qiagen, Germany, Cat. No. 69504), while DNA from fecal samples was extracted with the QIAamp® Fast DNA Stool Mini Kit (Qiagen, Canada, Cat. No. 51604). Where necessary, genomic DNA was fragmented using NEBNext® dsDNA Fragmentase (New England Biolabs, Cat. No. M0348S). Double-stranded libraries were prepared using the VAHTS Universal DNA Library Prep Kit for Illumina V4 (Vazyme, Cat. No. ND610). DNA libraries generated using museum skins and tissue samples were directly sequenced on the Illumina HiSeq X platform, producing 2 × 150-bp reads. For fecal-derived DNA libraries, we employed a capture hybridization approach prior to sequencing [18]. Biotin-labeled probes targeting the mitogenome were prepared via long-range PCR amplification of the human mitochondrial genome [19], purified with the Zymoclean™ Gel DNA Recovery Kit (Zymo; Cat. No. D4007) and biotinylated using the Biotin-Nick Translation mix (Roche; Cat. No. 11745824910). Libraries were incubated with the probes and enriched mitochondrial DNA was sequenced as described above.

Quality control and trimming of all raw data were conducted using fastp with the parameters -g -q 20 -u 50 -n 15 [20]. Mitogenomes were initially assembled and annotated using Mitoz v3.6 with default parameters [21]. For two samples of species *Moschus fuscus* (KIZ780414) and *Viverra megaspila* (KIZ650515) with relatively lower coverage of the mitogenomes, and the genomic Pacbio sequencing data of *Muntiacus vaginalis nigripes* we also aligned the reads to an

**Table 1. Species sequenced in this study.**

| Species | Source | voucher/isolate | Year | Sample locality |
|---|---|---|---|---|
| *Arctogalidia trivirgata* | Museum skin | KIZ830273 | 1971 | Xishuangbanna, Yunnan, China |
| *Arctogalidia trivirgata* | Museum skin | KIZ830274 | 1974 | Xishuangbanna, Yunnan, China |
| *Budorcas taxicolor whitei* | SRA | SRX15185605 | NA | NA |
| *Macaca munzala* | Feces | 223667 | 2023 | Lebugou, Tibet, China |
| *Moschus fuscus* | Museum skin | KIZ73444 | 1973 | Gongshan, Yunnan, China |
| *Moschus fuscus* | Museum skin | KIZ780414 | 1978 | Biluoxueshan, Yunnan, China |
| *Muntiacus vaginalis nigripes* | SRA | SRR28810463 | NA | NA |
| *Nomascus concolor* | Tissue | KIZ20090608 | 2009 | Yongdedaxueshan, Yunnan, China |
| *Nomascus hainanus* | Feces | MulanNH51 | 2022 | Hainan Tropical Rainforest National Park, Hainan, China |
| *Tragulus kanchil williamsoni* | Museum skin | KIZ75909 | 1959 | Mengla, Yunnan, China |
| *Tragulus kanchil williamsoni* | Museum skin | KIZ75910 | 1959 | Mengla, Yunnan, China |
| *Viverra megaspila* | Museum skin | KIZ650515 | 1975 | Chongzuo, Guangxi, China |

KIZ: Kunming Institute of Zoology, Chinese Academy of Sciences.

assembled mitogenome using BWA-MEM [22] and annotated using Geneious Prime® 2024.0.5 to cross verify the result. The raw genome-resequencing reads of *Budorcas taxicolor white* (SRX15185605) and *Muntiacus vaginalis nigripes* (SRR28810463) were downloaded from NCBI Sequence Read Archive (SRA) and processed similarly. All newly sequenced mitogenomes were verified manually and deposited in GenBank (Accession Numbers: PQ740948-PQ740958, BK070181).

### Curation of mammalian mitogenome reference sequences

To develop a curated mitochondrial genome dataset, we retrieved available mitogenome sequences from GenBank. Given the potential for misidentified submissions in GenBank, we employed the phylogenetic species concept (PSC) [23] to delineate species. Specifically, we evaluated whether sequences attributed to a given species formed a monophyletic clade. The most reliable sequence for each species was designated as its reference sequence.

To facilitate this analysis, we downloaded all available mitochondrial *CYTB* and *COI* gene sequences from GenBank for each order, as these genes are the most extensively represented in mammalian studies. All sequences were aligned with their corresponding mitogenomes using MAFFT v7.490 [24]. Maximum-likelihood (ML) phylogenetic trees were then constructed using IQ-TREE v1.6.8 [25] within PhyloSuite v1.2.2 [26]. For phylogenetic analyses, tRNA genes, the NADH dehydrogenase subunit 6 (*ND6)* gene, and the control region *(D-loop)* were excluded.

In most cases, each mitogenome clustered with its conspecific sequences with high bootstrap support (BS) (i.e., $BS \geq 80$), validating their taxonomic assignments. However, instances of paraphyly and polyphyly were observed in genera such as *Naemorhedus* and *Moschus*. For these cases, we examined sample localities and prioritized sequences from type localities as reference sequences. Pairwise genetic distances were calculated using the Kimura 2-parameter (K80) model, and genetic distance heatmaps were visualized using the R packages ape [27], seqinr [28], phangorn [29], and gplots (https://github.com/talgalili/gplots). Through this process, we established reliable reference mitogenome sequences for 164 species, ensuring accurate representation in our curated dataset.

## Results

### The updated list of state key protected wild mammals in China

The 2021 LSKPWA in China included 185 mammalian taxa. Following a comprehensive taxonomic revision and evaluation of species distribution, we determined that the updated list should include 169 valid species (S1 Table). This revision

involved the removal of eight taxa that are either confirmed to be absent from China's territory or have been extirpated in the wild within China. In addition, 11 taxa previously recognized as distinct species were reclassified as subspecies under known species. These include *Ovis ammon* (now including *collium*, *darwini*, *hodgsoni*, *karelini*, and *polii* as subspecies), *Cervus elaphus* (including *wallichii* and *yarkandensis* as subspecies), *Muntiacus vaginalis* (including *nigripes*), *Tragulus kanchil* (including *williamsoni*), *Budorcas tibetana* (including *bedfordi*) and *Budorcas taxicolor* (including *whitei*). Three taxa were synonymized with recognized species based on recent taxonomic evidence: *Delphinus capensis* was merged with *Delphinus delphis*; *Cervus canadensis* with *Cervus elaphus*; and *Naemorhedus griseus* with *Naemorhedus goral* [30,31]. Conversely, two former subspecies were elevated to full species status based on molecular and morphological studies: *A. styani*, previously considered a subspecies of *A. fulgens* [6] and *Trachypithecus melamera*, formerly treated as a subspecies of *Trachypithecus phayrei* (*T. phayrei shanicus*) [32]. In addition, we incorporated *Mesoplodon hotaula* (Deraniyagala's Beaked Whale) into the updated list following its first confirmed record in the South China Sea [33]. Finally, the pygmy slow loris *Nycticebus pygmaeus* has been reassigned to the newly established genus and species *Xanthonycticebus intermedius* [34,35].

## Phylogenetic relationships and species diversity

We successfully generated 12 new complete mitogenome sequences for eight species or subspecies either by using newly sequenced data (n = 10) or by extraction from the raw genome resequencing SRA reads (n = 2) (Table 1). Phylogenetic analyses of these new mitogenomes clustered them with their congeneric species validating their taxonomic classifications (Figs 1 and 2; S1 Fig). For example, the Hainan gibbon (*Nomascus hainanus*) and the Cao vit gibbon (*N. nasutus*) form sister taxa, occupying a basal position within the genus *Nomascus* (Fig 1A), being consistent with previous studies using *CYTB* gene [36]. The Arunachal macaque (*Macaca munzala*) was identified as the sister species to the *M. radiata* and *M. leucogenys* (BS = 100) (Fig 1B), corroborating earlier phylogenetic results based on *D-loop* and *CYTB* sequences [37,38]. The mitogenome of Large-spotted civet (*Viverra megaspila*; KIZ650515) formed a sister relationship with *V. zibetha* and *V. tangalunga* (Fig 1C), aligning with previous findings [39]. The two newly sequenced individual of the small-toothed palm civet (*Arctogalidia trivirgata*) clustered with a lineage previously identified in Myanmar (BS = 95), corresponding to the subspecies *A. trivirgata millsi* (Fig 1C). Two mitogenomes of *T. kanchil williamsoni* from Yunnan, China clustered within the clade of *T. kanchil* from Laos, affirming that *T. williamsoni* in China belongs to *T. kanchil* (Fig 1D). Similarly, the mitogenome of *Budorcas taxicolor whitei* clustered with other sequences of *B. taxicolor*, forming a well-supported sister lineage to the nominal subspecies *B. t. taxicolor* (BS = 100) (S1A Fig), being consistent with genomic evidence [40]. Finally, the mitogenome of *Muntiacus vaginalis nigripes* (SRR28810463), endemic to Hainan Island, clustered within *M. vaginalis* (BS = 100) (S1B Fig), supporting its treatment as a subspecies rather than a distinct species [41].

We observed paraphyly and polyphyly in finless porpoises (*Neophocaena*) and musk deer (*Moschus*). Certain clades of *Neophocaena* encompassed sequences labeled as different species (Fig 2A), indicating misidentification or the use of out-of-date taxonomy. Despite this, our phylogenetic tree aligns well with a previous study [42], suggesting an updated classification recognizing three distinct species: the Yangtze finless porpoise (*N. asiaeorientalis*), the East Asian finless porpoise (*N. sunameri*), and the Indo-Pacific finless porpoise (*N. phocaenoides*).

Within the genus *Moschus*, our phylogenetic analyses revealed several instances of taxonomic ambiguity (Fig 2B). One clade grouped the topotype of *M. anhuiensis* with sequences labeled as *M. berezovskii* and *M. chrysogaster* (BS = 100), exhibiting low genetic divergence (< 1.0%), suggesting potential misidentification or historical gene flow. *M. fuscus* appeared in two distinct clades: one included a sample from Gongshan (KIZ73444) (BS = 93), and the other comprised a sample from Biluoxueshan (KIZ780414), a sequence labeled as *M. leucogaster* from Tibet (NC_042604), and *M. chrysogaster from various localities* (BS = 97). Notably, NC_042604 did not cluster with the topotype of *M. leucogaster* from Nepal, indicating possible misidentification. Additionally, samples of *M. berezovskii* formed two genetically distinct

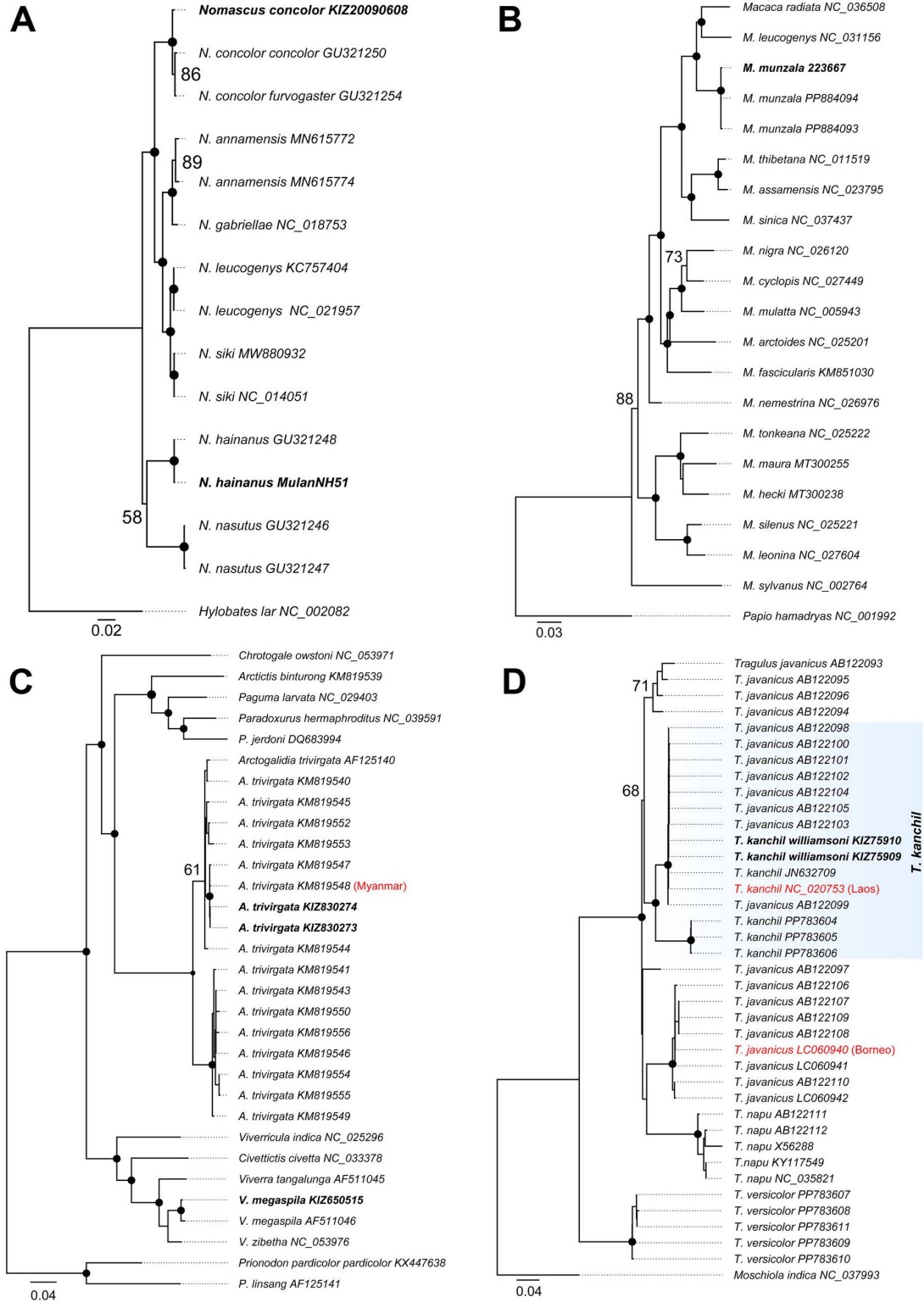

**Fig 1. The Maximum-likelihood (ML) phylogenetic tree of species newly sequenced in this study based on mitochondrial genome dataset.** (A) *Nomascus concolor* and *N. hainanus*; (B) *Macaca munzala*; (C) *Arctogalidia trivirgata* and *Viverra megaspila*. (D) *Tragulus kanchil williamsoni*. Nodes with BS ≥ 95 are indicated by black dots, while those with BS below 95 are shown as numbers. Bold names are newly obtained sequences in this study.

clades with a divergence of 5.9%: one from its type locality in Sichuan and another comprising subspecies *bijiangensis* and *caobangis* from Yunnan, suggesting unrecognized species-level diversity.

### A curated mitogenome database of state key protected wild mammals in China

Based on phylogenetic analyses, assessments of monophyly and type localities, we designated reference mitogenomes for 164 species. Mitogenomes were unavailable for five species including *M. peruvianus, M. hotaula, M. leucogaster, N. nasutus,* and *Ochotona iliensis*, for which only partial yet reliable *CYTB* and/or *D-loop* sequences are available ([S1 Table]).

## Discussion

This study enhances the taxonomic framework and mitogenomic resources for China's state key protected wild mammals. We revised the 2021 LSKPWA to reflect recent taxonomic updates, resulting in a refined list of 169 mammalian species that are protected. By integrating publicly available data with 12 newly sequenced mitogenomes, we designated curated reference sequences for 164 species. The resulting curated database provides a critical tool for resolving taxonomic ambiguities, clarifying phylogenetic relationships, and supporting forensic applications in wildlife conservation. Notably, seven of the 12 newly sequenced mitogenomes (58%) were derived from museum specimens, underscoring the enduring value of historical collections in modern conservation genomics [43], particularly for rare or elusive taxa such as the large-spotted civet, which was rediscovered after more than three decades [44].

### Taxonomic clarification and remaining challenges

Our findings clarified species boundaries in several cases but also highlighted ongoing challenges in taxonomic uncertainty. For example, the mouse-deer *Tragulus williamsoni* was recognized as a distinct species by Meijaard and Groves [45], but considered a subspecies of *T. kanchil* by others [7]. We sequenced two historical specimens (KIZ75909 and KIZ75910) from Mengla, Xishuangbanna, Yunnan, which clustered with *T. kanchil* from Laos, supporting its subspecific status. However, as the type locality of *williamsoni* is in Thailand, further confirmation through sequencing of topotype material remains essential.

Taxonomic complexity is even greater in the genus *Moschus*. Sequences labeled as *M. berezovskii* and *M. chrysogaster* appeared in multiple clades ([Fig 2B]), suggesting cryptic speciation, misidentification, or gene flow. For instance, *M. chrysogaster* (JQ608470) may represent a misidentified individual or a hybrid [46]. Similarly, the mitogenome of *M. leucogaster* (NC_042604) from Lazi County, Tibet, clusters closely with *M. fuscus*, rather than with *M. leucogaster* from its type locality in Nepal [47], indicating likely misidentification. Genetic divergence of 5.9% within *M. berezovskii,* exceeding typical intraspecific thresholds, further underscores the need for integrative approaches to reassess species boundaries [48]. Additionally, paraphyly within *M. fuscus* and *M. leucogaster* add to the uncertainty, raising the possibility of incomplete lineage sorting or past introgression. Collectively, these findings underscore the necessity for a comprehensive taxonomic reassessment of *Moschus*. Resolving such ambiguities will require integrative taxonomic approaches that combine mitogenomic data with nuclear genomic analyses, morphology, ecology, and geographic sampling.

### Forensic applications, future directions, and proposed best practices

The curated mitogenome database developed here, now encompassing 164 species and missing only five, represents a major step toward standardized species identification in China's protected mammals. By uniting updated taxonomy, curated reference sequences, and data from historical specimens, it provides an authoritative resource that can be directly

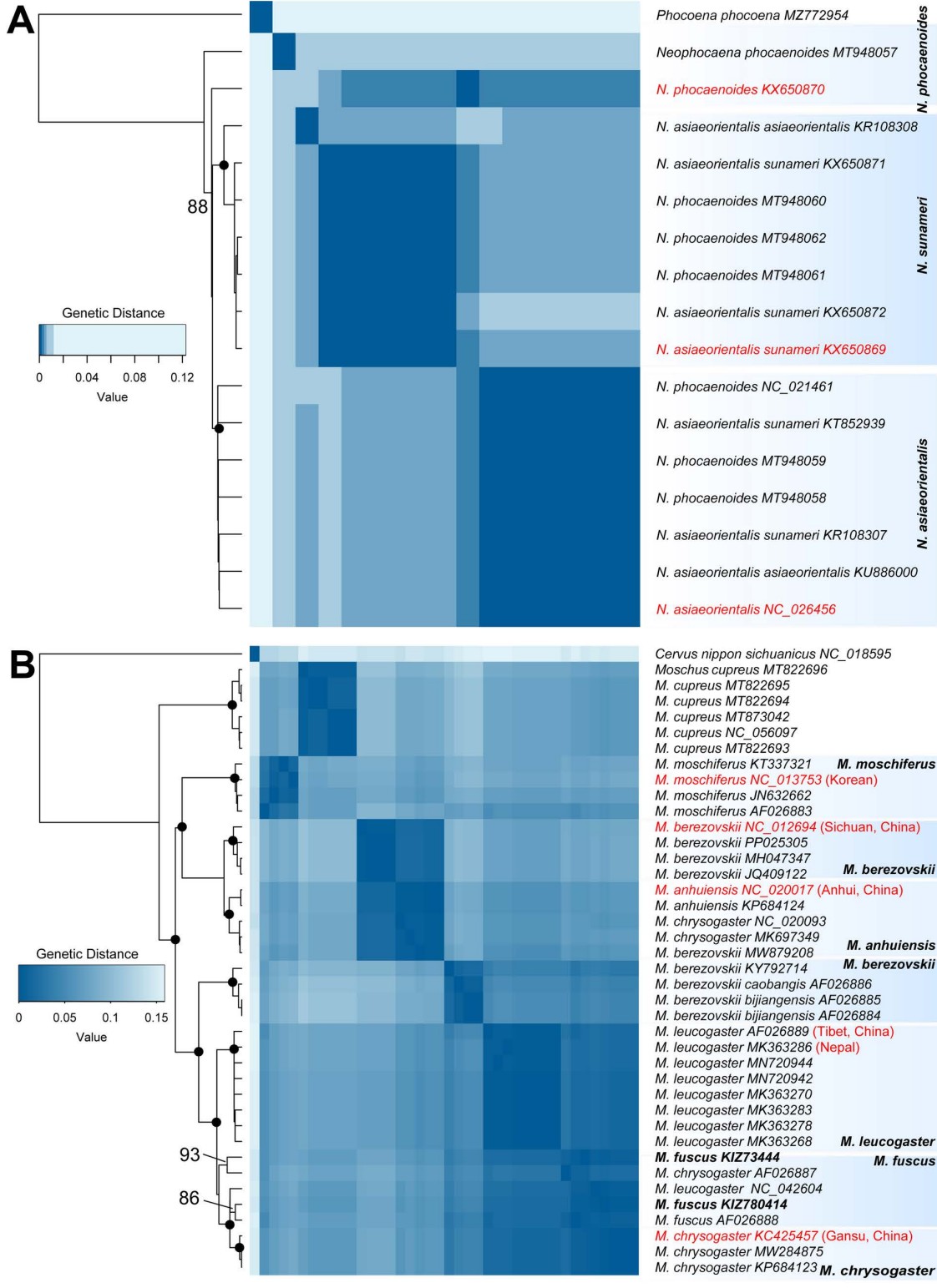

**Fig 2. Genetic distance heatmaps based on the *CYTB* gene using the Kimura 2-parameter (K80) model, integrated with ML phylogenetic trees constructed from the mitochondrial genome dataset.** (A) For the genus *Naemorhedus*, pairwise genetic distances between species range from 0.5% to 12.2%; (B) For the genus *Moschus*, pairwise genetic distances between species range from 0.9% to 15.6%. Nodes with BS ≥ 95 are indicated by black

dots, while those with BS below 95 are shown as numbers. Bold names are newly obtained sequences in this study. Red names are designated curated mitochondrial genome sequences for a species based on type locality and phylogenetic relationships. Sequences from GenBank used in this study are named with their accession numbers.

applied in biodiversity monitoring, forensic identification, and enforcement of conservation laws. However, the utility of this database will increase as it is expanded and refined, and its application should pioneer a more robust methodological standard for the field.

First, we strongly recommend the adoption of a phylogenetic framework for forensic species identification, rather than relying on simple genetic distance thresholds. As our results for *Moschus* demonstrate, genetic distance alone can be misleading in taxa with low interspecific divergence or complex evolutionary histories. Placing an unknown sample within a phylogenetic tree of curated reference sequences provides a far more robust and contextually supported identification. We propose that this phylogenetic approach should be established as a standard for forensic analysis, especially for legally protected species.

Second, our results demonstrate that reliance on single or geographically narrow reference sequences can be misleading for widespread species, in which intraspecific genetic structure can be substantial, and capturing this variation is critical. Incorporating mitogenomes from across the geographic ranges of widespread species will enable the assignment of confiscated specimens to specific populations or origins. Such resolution is vital for tracking illegal wildlife trade, as demonstrated in recent pangolin studies [49], and could be extended to other heavily trafficked groups such as musk deer and gorals.

Third, our database can serve as the foundation for expanding coverage to all mammals in China, not just those currently protected. Extending the database to species with ecological, economic, or cultural significance and eventually to all mammalian taxa would create a comprehensive national reference library. This expansion would enhance forensic capabilities by enabling identification of a broader range of trafficked species, including those not yet recognized as threatened but potentially at risk from overexploitation.

Fourth, our findings emphasize that the strength of forensic identification depends not only on genetic reference data but also on the robustness of the underlying taxonomy. In cases where taxonomy remains unresolved—such as *Moschus* and *Naemorhedus*—the reliability of forensic conclusions is diminished. Misaligned taxonomy can directly undermine the credibility of forensic reports in legal proceedings, highlighting the need for systematic taxonomic revision as a prerequisite for effective application of genetic tools. Moreover, in some groups, natural hybridization may occur, further complicating identification and requiring more nuanced frameworks for forensic reporting.

Finally, we acknowledge that mitochondrial genomes alone cannot fully resolve all cases of taxonomic or forensic ambiguity. Integration of nuclear and mitochondrial data, coupled with machine-learning approaches for species delimitation [48], which together will provide the most robust framework for both taxonomy and conservation enforcement.

## Supporting information

**S1 Fig. The Maximum-likelihood trees of *Budorcas taxicolor whitei* (A) and Muntiacus vaginalis nigripes (B) based on mitochondrial genome dataset.** Nodes with BS ≥ 95 are indicated by black dots, while those with BS below 95 are shown as numbers. Bold names are newly obtained sequences in this study. Red names are designated curated mitochondrial genome sequence for species based on type locality and phylogenetic relationships. Sequences from GenBank used in this study are named with their accession numbers.
(TIF)

**S1 Table. The updated list of State Key Protected Wild Mammals in China.**
(XLSX)

**S1 Dataset. The alignment of curated mitochondrial genome sequences is available online (**https://doi.org/10.57760/sciencedb.24175**) .**

(TXT)

## Acknowledgments

We are very grateful for the support of Kunming Natural History Museum of Zoology.

## Author contributions

**Conceptualization:** Song Li, Kai He.

**Data curation:** Xia Huang.

**Formal analysis:** Xia Huang, Guihua Zhang.

**Funding acquisition:** Kai He.

**Investigation:** Xia Huang, Guihua Zhang, Shiyun Liu, Shan Jiang, Yinhui Zhao, Pengfei Fan, Lianghua Huang, Xiaoming Zhang, Xueyou Li.

**Methodology:** Xia Huang, Guihua Zhang, Kai He.

**Project administration:** Kai He.

**Supervision:** Kai He.

**Validation:** Xia Huang.

**Visualization:** Xia Huang, Kai He.

**Writing – original draft:** Xia Huang, Kai He.

**Writing – review & editing:** Xia Huang, Joseph D. Orkin, Kai He.

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
