## [Decision Letter · Decision Letter 0]

13 Aug 2025

PONE-D-25-32177Curated Mitochondrial Genome Reference Database of State Key Protected Wild Mammal in ChinaPLOS ONE

Dear Dr. He,

Thank you for submitting your manuscript to PLOS ONE. After careful consideration, we feel that it has merit but does not fully meet PLOS ONE’s publication criteria as it currently stands. Therefore, we invite you to submit a revised version of the manuscript that addresses the points raised during the review process.

We look forward to receiving your revised manuscript.

Kind regards,

James Lee Crainey, Ph.D.

Academic Editor

PLOS ONE

Journal Requirements: 

3.  Please upload a copy of Supporting Information Table 1 which you refer to in your text on page 18.

Reviewers' comments:

Reviewer's Responses to Questions

**Comments to the Author**

1. Is the manuscript technically sound, and do the data support the conclusions?

Reviewer #1: Yes

Reviewer #2: Yes

2. Has the statistical analysis been performed appropriately and rigorously? 

Reviewer #1: Yes

Reviewer #2: Yes

3. Have the authors made all data underlying the findings in their manuscript fully available?

Reviewer #1: Yes

Reviewer #2: Yes

4. Is the manuscript presented in an intelligible fashion and written in standard English?

Reviewer #1: Yes

Reviewer #2: Yes

5. Review Comments to the Author

Reviewer #1: This study makes a highly valuable contribution to wildlife conservation genomics by establishing a curated mitochondrial genome reference database for China's state key protected wild mammals. One of its most commendable aspects is the integration of newly sequenced mitogenomes, especially from historical museum specimens, into a national-scale reference framework. This approach not only addresses critical taxonomic gaps but also enhances the practical utility of molecular tools in forensic identification and legal enforcement. The authors are encouraged to further expand this database in future work by including population-level variation to support more nuanced applications such as local provenance inference.

I just have some minor suggestions for this important work. While the manuscript is generally well written and scientifically sound, here are a few grammar and phrasing issues identified in the abstract and introduction:

While the study successfully curated mitogenomes for 164 mammalian species, most of the newly sequenced data were obtained from single individuals or specimens from a limited number of localities (e.g., museum skins, feces, or archived samples). This narrow sampling does not capture the potential intraspecific variation across different geographic regions within China, which is especially relevant for widespread or genetically structured species. If possible, authors should include a statement acknowledging the current limitation in geographic coverage and propose plans for future work that incorporate population-level sampling across the full distribution ranges of species.

Others:

1. changed "...resulting in a refined list of 169 mammalian species." To "...resulting in a refined list of 169 mammalian species that are protected."

2. Change "To capture nation-specific conservation need..." to "To capture nation-specific conservation needs..."

Reviewer #2: The manuscript presents a valuable and timely resource—a curated mitochondrial genome reference database for state-protected wild mammals in China. This dataset will support wildlife forensic applications, conservation genetics, and biodiversity monitoring. The combination of newly generated mitogenomes, historical museum specimens, and curated public databases demonstrates an integrated and innovative approach. The integration of updated taxonomic frameworks with genomic resources addresses a well-recognized gap in wildlife conservation and law enforcement, providing a reference standard for species identification. The successful generation of mitogenomes from archived museum specimens highlights the enduring value of natural history collections for genomic research. However, there are some issues to be resolved:

Major:

1. The updated species list (169 taxa) is based on taxonomic evidence; however, it would strengthen the paper if the decision-making framework (criteria for synonymization, elevation, or exclusion) were described in more detail.

2. The rationale for focusing solely on mitochondrial genomes (as opposed to including nuclear markers) is clear for forensic purposes, but could the authors briefly discuss limitations of mitogenomes for species delimitation, particularly in cases of introgression or hybridization?

3. The paper mentions taxonomic issues in Moschus and Naemorhedus but does not provide explicit recommendations for resolving these ambiguities (e.g., need for additional sampling or integration of nuclear genomes). Could the authors clarify how these findings should inform future taxonomic revisions?

4. In terms of data availability and usability, the curated dataset is said to support forensic applications. Are the curated sequences provided in a searchable database format (e.g., BLAST-enabled)? If not, are there plans to implement such an interface to improve usability?

Minor:

1. Line 163: The author provides the abbreviation of bootstrap value (BS), but BS should correspond to bootstrap rather than bootstrap value. However, the full name of a single word should be provided.

2. Figures 1 and 2 are informative, but some node labels and bootstrap values are difficult to read at the provided resolution. Consider enhancing the figure clarity or providing vector formats in the supporting information.

3. The term “Maximum-likelihood” appears in both legends of Figures 1 and 2, and is abbreviated as “ML” in the legend of Figure 2. Normally, the abbreviation should be given when the phrase appears for the first time.

4. Please use the recommended citation format of this journal.

6. PLOS authors have the option to publish the peer review history of their article (what does this mean? ). If published, this will include your full peer review and any attached files.

**Do you want your identity to be public for this peer review?** For information about this choice, including consent withdrawal, please see our Privacy Policy .

Reviewer #1: No

Reviewer #2: No

---

## [Author Response · Author response to Decision Letter 1]

25 Sep 2025

Thank you for the opportunity to revise our manuscript, “Curated Mitochondrial Genome Reference Database of State Key Protected Wild Mammal in China” (ID: PONE-D-25-32177) [EMID:0fe8ce408c6c3171]. We appreciate the valuable and constructive feedback provided by the reviewers, and we have thoroughly revised the manuscript to address all the points raised.

We have made substantial improvements throughout the paper, including the following key changes:

Abstract and Introduction: We revised these grammar and phrasing issues in this version according to one of the reviewer’s suggestions and have conducted a thorough check to ensure greater accuracy.

Materials and methods: This first paragraph of this section has been extensively rewritten to response one of the reviewer’s suggestions about the decision-making framework of taxonomic evidence. Each taxonomic revision was cross-referenced with the latest peer-reviewed publications and verified by relevant experts.

Abbreviation and Figures: We revised and conducted a thorough check for the abbreviation. To enhance the figure clarity, black dots were used to indicate nodes with bootstrap support (BS) ≥ 95, and the same base color with different transparency was employed to separate adjacent species.

Discussion: The discussion has been significantly expanded based on the reviewers' feedback. It now includes the need for broader geographic sampling as a key future direction, the inherent limitations of mitogenomic data, the integrative approach that includes nuclear genomics and advanced analytical methods to resolve the ambiguities in complex genera like Moschus and Naemorhedus. We recommend that accurate species identification should adopted a phylogenetic framework rather than based solely on genetic distance, particularly for taxa with low levels of intraspecific divergence.

References: We used the recommended citation format in this version.

We have provided a detailed, point-by-point response to each reviewer's comments in a separate document.

Finally, we wish to revise the financial disclosure. This work was funded by the National Natural Science Foundation of China, Grant numbers: U23A20161 and 32170452. The funders had no role in study design, data collection and analysis, decision to publish, or preparation of the manuscript.

We are confident that this revised manuscript represents a significant improvement.

---

## [Decision Letter · Decision Letter 1]

8 Oct 2025

Curated mitochondrial genome reference database of state key protected wild mammal in China

PONE-D-25-32177R1

Dear Dr. He,

We’re pleased to inform you that your manuscript has been judged scientifically suitable for publication and will be formally accepted for publication once it meets all outstanding technical requirements.

Kind regards,

James Lee Crainey, Ph.D.

Academic Editor

PLOS ONE

Reviewers' comments:

Reviewer's Responses to Questions

**Comments to the Author**

1. If the authors have adequately addressed your comments raised in a previous round of review and you feel that this manuscript is now acceptable for publication, you may indicate that here to bypass the “Comments to the Author” section, enter your conflict of interest statement in the “Confidential to Editor” section, and submit your "Accept" recommendation.

Reviewer #1: All comments have been addressed

Reviewer #2: All comments have been addressed

2. Is the manuscript technically sound, and do the data support the conclusions?

Reviewer #1: Yes

Reviewer #2: Yes

3. Has the statistical analysis been performed appropriately and rigorously? 

Reviewer #1: Yes

Reviewer #2: Yes

4. Have the authors made all data underlying the findings in their manuscript fully available?

Reviewer #1: Yes

Reviewer #2: Yes

5. Is the manuscript presented in an intelligible fashion and written in standard English?

Reviewer #1: Yes

Reviewer #2: Yes

6. Review Comments to the Author

Reviewer #1: All comments are well-revised. The presented version with high-quality design and writing, which is being worth published.

Reviewer #2: (No Response)

7. PLOS authors have the option to publish the peer review history of their article (what does this mean? ). If published, this will include your full peer review and any attached files.

**Do you want your identity to be public for this peer review?** For information about this choice, including consent withdrawal, please see our Privacy Policy .

Reviewer #1: No

Reviewer #2: No

---

## [Editor Report · Acceptance letter]

PONE-D-25-32177R1

PLOS ONE

Dear Dr. He,

I'm pleased to inform you that your manuscript has been deemed suitable for publication in PLOS ONE. Congratulations! Your manuscript is now being handed over to our production team.

Kind regards,

on behalf of

Dr. James Lee Crainey

Academic Editor

PLOS ONE